# Online Bayesian Goal Inference for Boundedly-Rational Planning Agents

**Tan Zhi-Xuan,    Jordyn L. Mann,    Tom Silver**
**Joshua B. Tenenbaum,     Vikash K. Mansinghka**
Massachusetts Institute of Technology
{xuan,jordynm,tslvr,jbt,vkm}@mit.edu

## Abstract

People routinely infer the goals of others by observing their actions over time. Remarkably, we can do so even when those actions lead to failure, enabling us to assist others when we detect that they might not achieve their goals. How might we endow machines with similar capabilities? Here we present an architecture capable of inferring an agent's goals online from both optimal and non-optimal sequences of actions. Our architecture models agents as boundedly-rational planners that interleave search with execution by replanning, thereby accounting for sub-optimal behavior. These models are specified as probabilistic programs, allowing us to represent and perform efficient Bayesian inference over an agent's goals and internal planning processes. To perform such inference, we develop Sequential Inverse Plan Search (SIPS), a sequential Monte Carlo algorithm that exploits the online replanning assumption of these models, limiting computation by incrementally extending inferred plans as new actions are observed. We present experiments showing that this modeling and inference architecture outperforms Bayesian inverse reinforcement learning baselines, accurately inferring goals from both optimal and non-optimal trajectories involving failure and back-tracking, while generalizing across domains with compositional structure and sparse rewards.

## 1 Introduction

Everyday experience tells us that it is impossible to plan ahead for everything. Yet, not only do humans still manage to achieve our goals by piecing together partial and approximate plans, we also appear to account for this cognitive strategy when inferring the goals of others, understanding that they might plan and act sub-optimally, or even fail to achieve their goals. Indeed, even 18-month old infants seem capable of such inferences, offering their assistance to adults after observing them execute failed plans [1]. How might we understand this ability to infer goals from such plans? And how might we endow machines with this capacity, so they might assist us when our plans fail?

While there has been considerable work on inferring the goals and desires of agents, much of this work has assumed that agents act optimally to achieve their goals. Even when this assumption is relaxed, the forms of sub-optimality considered are often highly simplified. In inverse reinforcement learning, for example, agents are assumed to either act optimally [2] or to exhibit Boltzmann-rational action noise [3], while in the plan recognition literature, longer plans are assigned exponentially decreasing probability [4]. None of these approaches account for the difficulty of planning itself, which may lead agents to produce sub-optimal or failed plans. This not only makes them ill-equipped to infer goals from such plans, but also saddles them with a cognitively implausible burden: If inferring an agent's goals requires knowing the optimal solution to reach each goal, then an observer would need to compute the optimal plan or policy for *all* of those goals in advance [5]. Outside of the simplest problems and domains, this is deeply intractable.

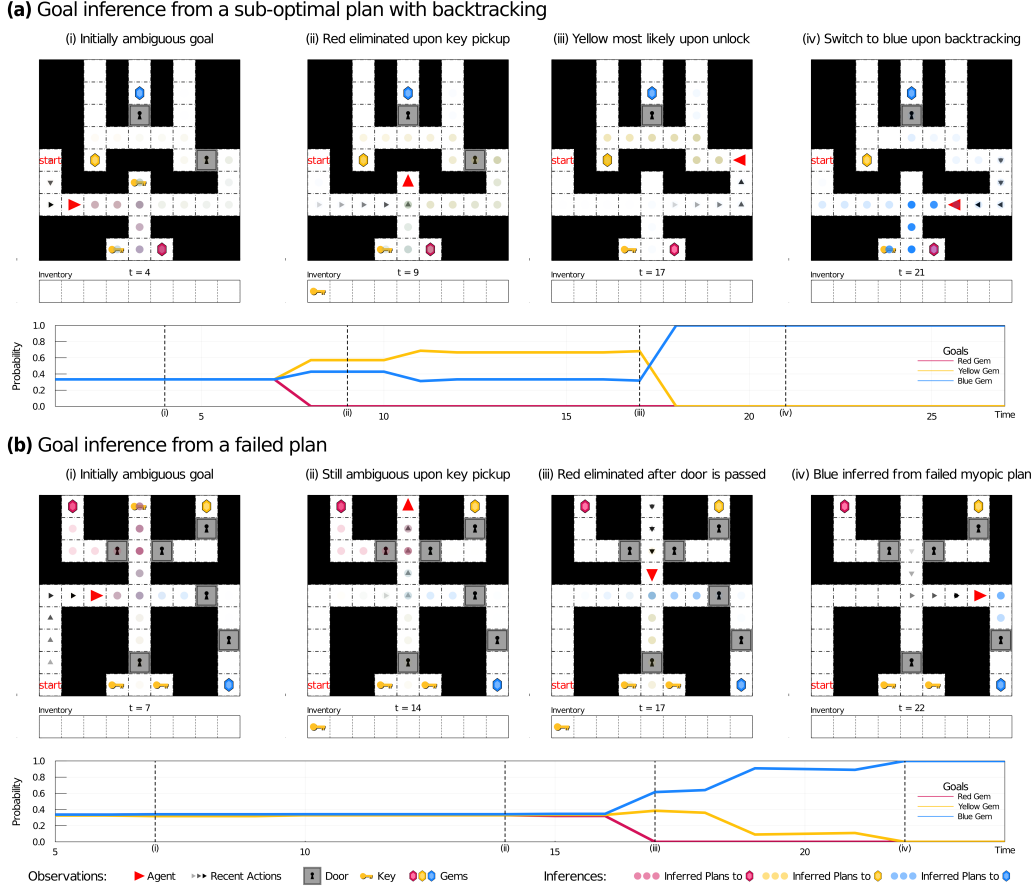

**(a)** Goal inference from a sub-optimal plan with backtracking

(i) Initially ambiguous goal  (ii) Red eliminated upon key pickup  (iii) Yellow most likely upon unlock  (iv) Switch to blue upon backtracking

**(b)** Goal inference from a failed plan

(i) Initially ambiguous goal  (ii) Still ambiguous upon key pickup  (iii) Red eliminated after door is passed  (iv) Blue inferred from failed myopic plan

Observations: ▶ Agent  ▸▸▸ Recent Actions  ▣ Door  ⚷ Key  ●●● Gems     Inferences: ●●● Inferred Plans to ●   ●●● Inferred Plans to ●   ●●● Inferred Plans to ●

**Figure 1:** Our architecture performing online Bayesian goal inference via Sequential Inverse Plan Search. In **(a)**, an agent exhibits a *sub-optimal plan* to acquire the blue gem, backtracking to pick up the key required for the second door. In **(b)**, an agent exhibits a *failed plan* to acquire the blue gem, myopically using up its first key to get closer to the gem instead of realizing that it needs to collect the bottom two keys. In both cases, our method not only manages to infer the correct goal by the end, but also captures sharp human-like shifts in its inferences at key points, such as **(a.ii)** when the agent picks up a key unnecessary for the red gem, **(a.ii)** when the agent starts to backtrack, **(b.iii)** when the agent ignores the door to the red gem, or **(b.iv)** when the agent unlocks the first door to the blue gem.

In this paper, we present a unified modeling and inference architecture (Figure 2) that addresses both of these limitations. In contrast to prior work that models agents as *actors* that are *noisily rational*, we model agents as *planners* that are *boundedly rational* with respect to how much they plan, interleaving resource-limited plan search with plan execution. This allows us to perform online Bayesian inference of plans and goals even from highly sub-optimal trajectories involving backtracking or irreversible failure (Figure 1). We do so by modeling agents as probabilistic programs (Figure 3), comprised of goal priors and domain-general planning algorithms (Figure 2i), and interacting with a symbolic environment model (Figure 2ii). Inference is then performed via Sequential Inverse Plan Search (SIPS), a sequential Monte Carlo (SMC) algorithm that exploits the replanning assumption of our agent models, incrementally inferring partial plans while limiting computational cost (Figure 2iii).

Our architecture delivers both accuracy and speed by being built in Gen, a general-purpose probabilistic programming system that supports customized inference using data-driven proposals and involutive rejuvenation kernels [6, 7, 8], alongside an embedding of the Planning Domain Definition Language [9, 10], enabling the use of fast general-purpose planners [11] as modeling components. We evaluate our approach against a Bayesian inverse reinforcement learning baseline [12] on a wide variety of planning domains that exhibit compositional task structure and sparse rewards (e.g. Figure 1), achieving high accuracy on many domains, often with orders of magnitude less computation.

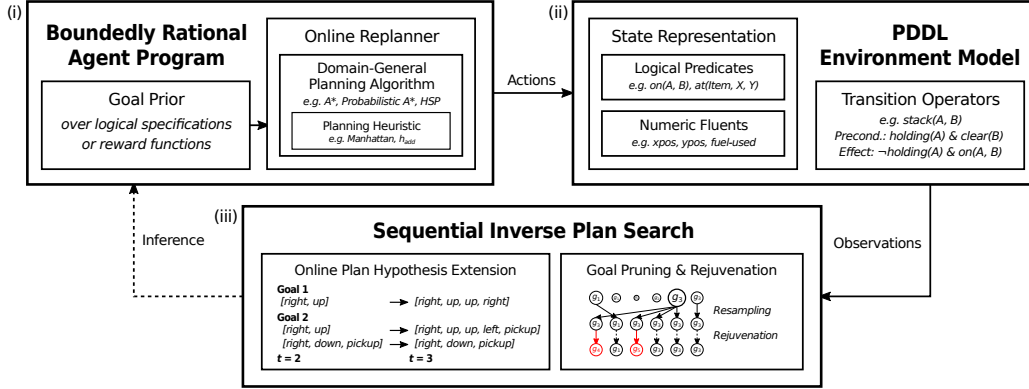

**Figure 2:** Our modeling and inference architecture is comprised of: **(i)** A programmatic model of a boundedly rational planning agent, implemented in the Gen probabilistic programming system; **(ii)** An environment model specified in the Planning Domain Definition Language (PDDL), facilitating support for a wide variety of planning domains and state-of-the-art symbolic planners; **(iii)** Sequential Inverse Plan Search (SIPS), a novel SMC algorithm that exploits the replanning assumption of our agent model to reduce computation, extending hypothesized plans only as new observations arrive.

## 2   Related Work

**Inverse reinforcement learning (IRL).** A long line of work has shown how to learn reward functions as explanations of goal-directed agent behavior via inverse reinforcement learning [2, 13, 12, 14]. However, most such approaches are too costly for online settings of complex domains, as they require solving the underlying Markov Decision Process (MDP) for every posited goal or reward function, and for all possible initial states [15, 5]. Our approach instead assumes that agents are online model-based planners. This greatly reduces computation time, while also better reflecting humans' intuitive understanding of other agents.

**Bayesian theory-of-mind (BToM).** Computational models of humans' intuitive theory-of-mind posit that we understand other's actions by Bayesian inference of their likely goals and beliefs. These models, largely built upon the same MDP formalism used in IRL, have been shown to make predictions that correspond closely with human inferences [16, 17, 18, 19, 20, 21, 22]. Some recent work also models agents using probabilistic programs [23, 24]. Our research extends this line of work by explicitly modeling an agent's partial plans, or *intentions* [25]. This allows our architecture to infer final goals from instrumental subgoals produced as part of a plan, and to account for sub-optimality in those plans, thereby enriching the range of mental inferences that BToM models can explain.

**Plan recognition as planning (PRP).** Our work is related to the literature on plan recognition as planning, which performs goal and plan inference by using classical satisficing planners to model plan likelihoods given a goal [26, 4, 27, 28, 29, 30]. However, because these approaches use a heuristic likelihood model that assumes goals are always achievable, they are unable to infer likely goals when irreversible failures occur. In contrast, we model agents as online planners who may occasionally execute partial plans that lead to dead ends.

**Online goal inference.** Several recent papers have extended IRL to an online setting, but these have either focused on maximum-likelihood estimation in 1D state spaces [31, 32], or utilize an expensive value iteration subroutine that is unlikely to scale [33]. In contrast, we develop a sequential Monte Carlo algorithm that exploits the online nature of the agent models in order to perform incremental plan inference with limited computation cost.

**Inferences from sub-optimal behavior.** We build upon a growing body of research on inferring goals and preferences while accounting for human sub-optimality [3, 24, 34, 35, 36, 37], introducing a model of boundedly-rational planning as resource-limited search. This reflects a natural principle of *resource rationality* under which agents are less likely to engage in costly computations [38, 39]. Unlike prior models of myopic agents which assign zero reward to future states beyond some time horizon [34, 36], our approach accounts for myopic planning in domains with instrumental subgoals and sparse rewards.

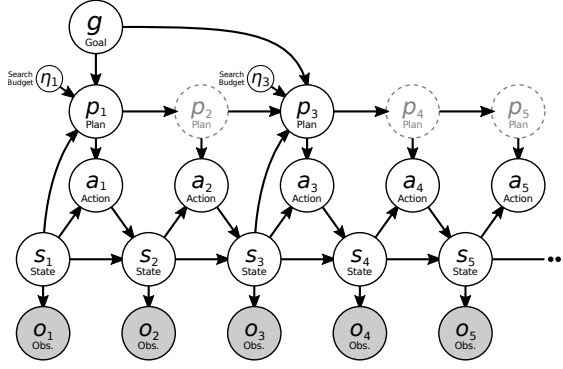

```
model UPDATE-PLAN(t, s_t, p_{t-1}, g)
    parameters: PLANNER, r, q, γ, h
    if t > LENGTH(p_{t-1}) or s_t ∉ p_{t-1}[t] then
        η ~ NEGATIVE-BINOMIAL(r, q)
        p̃_t ~ PLANNER(s_t, g, h, γ, η)
        p_t ← APPEND(p_{t-1}, p̃_t)
    else
        p_t ← p_{t-1}
    end if
    return p_t
end model
        (i) Samples from P(p_t|s_t, p_{t-1}, g)

model SELECT-ACTION(t, s_t, p_t)
    return p_t[t][s_t]
end model
        (ii) Samples from P(a_t|s_t, p_t)
```

**(a)** One realization of our agent and environment model.      **(b)** Boundedly-rational agent programs.

**Figure 3:** We model agents as boundedly rational planners that interleave search and execution of partial plans as they interact with the environment. In **(a)** we depict one possible realization of this model, where the agent initially samples a search budget $\eta_1$ and searches for a plan $p_1$ that is two actions long. At $t = 2$, no additional planning needs to be done, so $p_2$ is copied from $p_1$, as denoted by the dashed lines. The agent then replans at $t = 3$ from state $s_3$, sampling a new search budget $\eta_3$ and an extended plan $p_3$ with three more actions. We formally specify this agent model using probabilistic programs, with pseudo-code shown in **(b)**. UPDATE-PLAN samples extended plans $p_t$ given previous plans $p_{t-1}$, while SELECT-ACTION selects an action $a_t$ according the current plan $p_t$.

## 3 Boundedly-Rational Planning Agents

In order to account for sub-optimal behavior due to resource-limited planning, observers need to model not only an agent's goals and actions, but also the plans they form to achieve those goals. As such, we model agents and their environments as generative processes of the following form:

$$
\begin{align}
\textit{Goal prior:} \quad & g \sim P(g) \tag{1}\\
\textit{Plan update:} \quad & p_t \sim P(p_t|s_t, p_{t-1}, g) \tag{2}\\
\textit{Action selection:} \quad & a_t \sim P(a_t|s_t, p_t) \tag{3}\\
\textit{State transition:} \quad & s_{t+1} \sim P(s_{t+1}|s_t, a_t) \tag{4}\\
\textit{Observation noise:} \quad & o_{t+1} \sim P(o_{t+1}|s_{t+1}) \tag{5}
\end{align}
$$

where $g$, $p_t$, $a_t$, $s_t$ are the agent's goals, the internal state of the agent's plan, the agent's action, and the environment's state at time $t$ respectively. For the purposes of goal inference, observers also assume that each state $s_t$ might be subject to observation noise, producing an observed state $o_t$.

This generative process, depicted in Figure 3a, extends the standard model of MDP agents by modeling plans and plan updates explicitly, allowing us to represent not only agents that act according to some precomputed policy $a_t \sim \pi(a_t|s_t)$, but also agents that compute and update their plans $p_t$ on-the-fly. We describe each component of this process in greater detail below.

### 3.1 Modeling Goals, States and Observations

To represent states, observations, goals, and distributions over goals in a general and flexible manner, our architecture embeds the Planning Domain Definition Language (PDDL) [9, 10], representing states $s_t$ and goals $g$ in terms of predicate-based facts, relations, and numeric expressions (Figure 2ii). State transitions $P(s_t|s_{t-1}, a_{t-1})$ are modeled by transition operators that specify the preconditions and effects of actions. While we focus on deterministic transitions in this paper, we also support stochastic transitions, as in Probabilistic PDDL [40]. Given this representation, an observer's prior over goals $P(g)$ can be specified as a probabilistic program over PDDL goal specifications, including numeric reward functions, as well as sets of goal predicates (e.g. `has(gem)`), equivalent to indicator reward functions. Observation noise $P(o_{t+1}|s_{t+1})$ can also be modeled by corrupting each Boolean predicate with some probability, and adding continuous (e.g. Gaussian) noise to numeric fluents.

## 3.2 Modeling Sub-Optimal Plans and Actions

To model sub-optimal plans, the basic insight we follow is that agents like ourselves are *boundedly rational*: we *attempt* to plan to achieve our goals efficiently, but are limited by our cognitive resources. The primary limitation we consider is that full-horizon planning is often costly or intractable. Instead, it may often make sense to form partial plans towards promising intermediate states, execute them, and replan from there. We model this by assuming that agents only search for a plan up to some budget $\eta$, before executing a partial plan to a promising state found during search. We operationalize $\eta$ as the maximum number of nodes expanded (i.e., states explored), which we treat as a random variable sampled from a negative binomial distribution:

$$\eta \sim \text{NEGATIVE-BINOMIAL}(r, q) \tag{6}$$

The parameters $r$ (maximum failure count) and $q$ (continuation probability) characterize the persistence of a planner who may choose to give up after expanding each node. When $r > 1$, this distribution peaks at medium values of $\eta$, then decreases exponentially, modeling agents that are unlikely to form extremely long plans, which are costly, or extremely short plans, which are unhelpful.

This model also assumes access to a planning algorithm capable of producing partial plans. While we support *any* such planner as a sub-component, in this work we focus on A* search due to its ability to support domain-general heuristics that can guide search in human-like ways [11, 41]. We also modify A* so that search is stochastic, modeling agent sub-optimality during search. In particular, instead of always expanding the most promising successor state, we sample successor $s$ with probability:

$$P_{\text{expand}}(s) \propto \exp(-f(s, g)/\gamma) \tag{7}$$

where $\gamma$ is a noise parameter controlling the randomness of search, and $f(s, g) = c(s) + h(s, g)$ is the estimated total plan cost, i.e. the sum of the path cost $c(s)$ so far with the heuristic goal distance $h(s, g)$. On termination, we simply return the most recently selected successor state, which is likely to have low total plan cost $f(s, g)$ if the heuristic $h(s, g)$ is informative and the noise $\gamma$ is low.

We incorporate these limitations into a model of how a boundedly rational planning agent interleaves search and execution, specified by the probabilistic programs UPDATE-PLAN and SELECT-ACTION in Figure 3b. At each time $t$, the agent may reach the end of its last made plan $p_{t-1}$ or encounter a state $s_t$ not anticipated by the plan, in which case it will call the base planner (probabilistic A*) with a randomly sampled node budget $\eta$. The partial plan produced is then used to extend the original plan. Otherwise, the agent will simply continue executing its original plan, performing no additional computation. Note that by replanning when the unexpected occurs, the agent automatically handles some amount of stochasticity, as well as errors in its environment model.

## 4 Online Bayesian Goal Inference

Having specified our model, we can now state the problem of Bayesian goal inference. We assume that an observer receives a sequence of potentially noisy state observations $o_{1:t} = (o_1, ..., o_t)$. Given the observations up to timestep $t$ and a set of possible goals $\mathcal{G}$, the observer's aim is to infer the agent's goal $g \in \mathcal{G}$ by computing the posterior:

$$P(g|o_{1:t}) \propto P(g) \sum_{\substack{s_{1:t} \\ a_{1:t} \\ p_{1:t}}} \prod_{\tau=0}^{t-1} P(o_{\tau+1}|s_{\tau+1})P(s_{\tau+1}|s_\tau, a_\tau)P(a_\tau|s_\tau, p_\tau)P(p_\tau|s_\tau, p_{\tau-1}, g) \tag{8}$$

Computing this posterior exactly is intractable, as it requires marginalizing over all the random latent variables $s_\tau$, $a_\tau$, and $p_\tau$. Instead, we develop a sequential Monte Carlo procedure, shown in Algorithm 1, to perform approximate inference in an online manner, using samples from the posterior $P(g|o_{1:t-1})$ at time $t-1$ to inform sampling from the posterior $P(g|o_{1:t})$ at time $t$. We call this algorithm Sequential Inverse Plan Search (SIPS), because it sequentially inverts a search-based planning algorithm, inferring sequences of partial plans that are likely given the observations, and consequently the likely goals.

As in standard particle filtering schemes, we first sample a set of particles or hypotheses $i \in [1, k]$, with corresponding weights $w_i$ (lines 3-5). Each particle corresponds to a particular plan $p_\tau^i$ and goal $g^i$. As each new observation $o_\tau$ arrives, we extend the particles (lines 12–14) and reweight them by their likelihood of producing that observation (line 15). The collection of weighted particles thus approximates the full posterior over the unobserved variables in our model, including the agent's plans and goals. We describe several key features of this algorithm below.

---

**Algorithm 1** Sequential Inverse Plan Search (SIPS) for online Bayesian goal inference

---

1: **procedure** SIPS($s_0, o_{1:t},$ )
2:     **parameters:** $k$, number of particles; $c$, resampling threshold
3:     $w^i \leftarrow 1$ **for** $i \in [1, k]$                                      ▷ Initialize particle weights
4:     $s_0^i, p_0^i, a_0^i \leftarrow s_0, [], \text{no-op}$ **for** $i \in [1, k]$            ▷ Initialize states, plans and actions
5:     $g^i \sim$ GOAL-PRIOR() **for** $i \in [1, k]$              ▷ Sample $k$ particles from goal prior
6:     **for** $\tau \in [1, t]$ **do**
7:         **if** EFFECTIVE-SAMPLE-SIZE($w^1, ..., w^k$)/$k < c$ **then**       ▷ Resample and rejuvenate
8:             $g^i, s_{1:\tau}^i, p_{1:\tau}^i, a_{1:\tau}^i \sim$ RESAMPLE($[g^i, s_{1:\tau}, p_{1:\tau}, a_{1:\tau}]^{1:k}$) **for** $i \in [1, k]$
9:             $g^i, s_{1:\tau}^i, p_{1:\tau}^i, a_{1:\tau}^i \sim$ REJUVENATE($g^i, o_{1:\tau}, s_{1:\tau}^i, p_{1:\tau}^i, a_{1:\tau}^i$) **for** $i \in [1, k]$
10:         **end if**
11:         **for** $i \in [1, k]$ **do**                    ▷ Extend each particle to timestep $\tau$
12:             $s_\tau^i \sim P(s_\tau | s_{\tau-1}^i, a_{\tau-1}^i)$               ▷ Sample state transition
13:             $p_\tau^i \sim$ UPDATE-PLAN($p_\tau | s_\tau^i, p_{\tau-1}^i, g^i$)         ▷ Extend plan if necessary
14:             $a_\tau^i \sim$ SELECT-ACTION($a_\tau | s_\tau^i, p_\tau^i$)              ▷ Select action
15:             $w^i \leftarrow w^i \cdot P(o_\tau | s_\tau^i)$                 ▷ Update particle weight
16:         **end for**
17:     **end for**
18:     $\tilde{w}^i \leftarrow w^i / \sum_{j=1}^k w^j$ **for** $i \in [1, k]$            ▷ Normalize particle weights
19:     **return** $[(g^1, w^1), ..., (g^k, w^k)]$              ▷ Return weighted goal particles
20: **end procedure**
21:
22: **procedure** REJUVENATE($g, o_{1:\tau}, s_{1:\tau}, p_{1:\tau}, a_{1:\tau}$)       ▷ Metropolis-Hasting rejuvenation move
23:     **parameters:** $p_g$, goal rejuvenation probability
24:     **if** BERNOULLI($p_g$) **then**                       ▷ Heuristic-driven goal proposal
25:         $g' \sim Q(g) := $ SOFTMAX($[h(o_\tau, g)$ for $g \in \mathcal{G}]$)   ▷ Propose $g_0'$ based on est. distance to $o_\tau$
26:         $s_{1:\tau}', p_{1:\tau}', a_{1:\tau}' \sim P(s_{1:\tau}, p_{1:\tau}, a_{1:\tau} | g)$        ▷ Sample trajectory under new goal $g$
27:         $\alpha \leftarrow Q(g)/Q(g')$                            ▷ Compute proposal ratio
28:     **else**                                  ▷ Error-driven replanning proposal
29:         $t_* \sim Q(t_* | s_{1:\tau}, o_{1:\tau})$          ▷ Sample a time close to when $s_{1:\tau}$ diverges from $o_{1:\tau}$
30:         $s_{t_*:\tau}', p_{t_*:\tau}', a_{t_*:\tau}' \sim Q(s_{t_*:\tau}, p_{t_*:\tau}, a_{t_*:\tau} | o_{t_*:\tau})$    ▷ Propose new plan sequence $p_{t_*:\tau}'$
31:         $\alpha \leftarrow Q(s_{t_*:\tau}, p_{t_*:\tau}, a_{t_*:\tau} | o_{t_*:\tau})/Q(s_{t_*:\tau}', p_{t_*:\tau}', a_{t_*:\tau}' | o_{t_*:\tau})$   ▷ Compute proposal ratio
32:         $\alpha \leftarrow \alpha \cdot Q(t_* | s_{1:\tau}', o_{1:\tau})/Q(t_* | s_{1:\tau}, o_{1:\tau})$       ▷ Reweight by auxiliary proposal ratio
33:     **end if**
34:     $\alpha \leftarrow \alpha \cdot P(o_{1:\tau} | s_{1:\tau}')/P(o_{1:\tau} | s_{1:\tau})$              ▷ Compute acceptance ratio
35:     **return** $g_0', s_{1:\tau}', p_{1:\tau}', a_{1:\tau}'$ **if** BERNOULLI(min($\alpha, 1$)) **else** $g_0, s_{1:\tau}, p_{1:\tau}, a_{1:\tau}$   ▷ Accept or reject proposals
36: **end procedure**

---

## 4.1 Online Extension of Hypothesized Partial Plans

A key aspect that makes SIPS a genuinely *online* algorithm is the modeling assumption that agents also plan *online*. This obviates the need for the observer to precompute a complete plan or policy for each of the agent's possible goals in advance, and instead defers such computation to the point where the agent reaches a time $t$ that the observer's hypothesized plans do not yet reach. In particular, for each particle $i$, the corresponding plan hypothesis $p_{t-1}^i$ is extended (Algorithm 1, line 13) by running the UPDATE-PLAN procedure in Figure 3b.i, which only performs additional computation if $p_{t-1}^i$ does not already contain a planned action for time $t$ and state $s_t$. This means that at any given time $t$, only a small number of plans require extension, limiting the number of expensive planning calls.

## 4.2 Managing Hypothesis Diversity via Resampling and Rejuvenation

We also introduce resampling and rejuvenation steps into SIPS in order to ensure particle diversity. Whenever the effective sample size falls below a threshold $c$ (line 7), we resample the particles (line 8), thereby pruning low-weight hypotheses. We then rejuvenate by applying a mixture of two data-driven Metropolis-Hastings kernels to each particle. The first kernel uses a heuristic-driven goal proposal (lines 25-27), proposing goals $\tilde{g} \in \mathcal{G}$ which are close in heuristic distance $h(o_\tau, \tilde{g})$ to the last observed state $o_\tau$. This allows SIPS to reintroduce goals that were pruned, but later become more likely. The second kernel uses an error-driven replanning proposal (lines 29-32), which samples a time close to the divergence point between the hypothesized and observed trajectories, and then proposes to replan from that time, thereby constructing a new sequence of hypothesized partial plans that are less likely to diverge from the observations. Despite the complexity of these proposals, acceptance ratios are automatically calculated via Gen's support for involutive kernels [8]. Collectively, these steps help to ensure that hypotheses are both diverse and likely given the observations.

# 5 Experiments

We conducted several sets of experiments that demonstrate the human-likeness, accuracy, speed, and robustness of our approach. We first present experiments demonstrating the novel capacity of SIPS to infer goals from sub-optimal trajectories involving backtracking and failure (Figure 1). Comparing these inferences against human goal inferences shows that SIPS is more human-like than baseline approaches (Figure 4). We also evaluate the accuracy and speed of SIPS on a variety of planning domains (Table 1a), showing that it outperforms Bayesian IRL baselines. Finally, we present robustness experiments showing that SIPS can infer goals even when the data-generating model differs from the model assumed by the algorithm (Table 1b).

## 5.1 Domains

We validate our approach on domains with varying degrees of complexity, both in terms of the size of the state space $|\mathcal{S}|$ and the number of possible goals $|\mathcal{G}|$. All domains are characterized by compositional structure and sparse rewards, posing a challenge for standard MDP-based approaches.

**Taxi** ($|\mathcal{G}| = 3, |\mathcal{S}| = 125$): A benchmark domain used in hierarchical reinforcement learning [42], where a taxi has to transport a passenger from one location to another in a gridworld.

**Doors, Keys, & Gems** ($|\mathcal{G}| = 3, |\mathcal{S}| \sim 10^5$): A domain in which an agent must navigate a maze with doors, keys, and gems (Figure 1). Each key can be used once to unlock a door, allowing the agent to acquire items behind that door. Goals correspond to acquiring one out of three colored gems.

**Block Words** ($|\mathcal{G}| = 5, |\mathcal{S}| \sim 10^5$): A Blocks World variant adapted from [4] where blocks are labeled with letters. Goals correspond to block towers that spell one of a set of five English words.

**Intrusion Detection** ($|\mathcal{G}| = 20, |\mathcal{S}| \sim 10^{30}$): A cybersecurity-inspired domain drawn from [4], where an agent might perform a variety of attacks on a set of servers. There are 20 possible goals, each corresponding to a set of attacks (e.g. cyber-vandalism or data-theft) on up to 10 servers.

## 5.2 Baselines

We implemented Bayesian IRL (BIRL) baselines by running value iteration to compute a Boltzman-rational policy $\pi(a_t|s_t, g)$ for each possible goal $g \in \mathcal{G}$. Following the setting of early Bayesian theory-of-mind approaches [18], we treated goals as indicator reward functions, and assumed a uniform prior $P(g)$ over goals. Inference was then performed by exact computation of the posterior over reward functions, using the policy as the likelihood for observed actions. Unless otherwise stated, we used a discount factor of 0.9, and Boltzmann noise parameter $\alpha$=1.

Due to the exponentially large state space of many of our domains, standard value iteration (VI) often failed to converge even after $10^6$ iterations. As such, we implemented two variants of BIRL that use asynchronous VI, sampling states instead of fully enumerating them. The first, *unbiased BIRL*, uses uniform random sampling of the state space up to 250,000 iterations, sufficient for convergence in the Block Words domain. The second, *oracle BIRL*, assumes oracular access to the full set of observed trajectories in advance, and performing biased sampling of states that appear in those trajectories. Although inapplicable in practice for online use, this ensures that the computed policy is able to reach the goal in all cases, making it a useful benchmark for comparison.

## 5.3 Human-Like Goal Inference from Sub-optimal and Failed Plans

To investigate the novel human-like capabilities of our approach, we performed a set of qualitative experiments on a set of trajectories designed to exhibit notable sub-optimality or failure. The experiments were performed on the Doors, Keys & Gems domain because it allows for irreversible failures. Two illustrative examples are shown in Figure 1, and more are provided in the supplement. In Figure 1a, SIPS accurately infers goals from a sub-optimal plan with backtracking, initially placing more posterior mass on the yellow gem when the agent acquires the first key (panel ii), but then switching to the blue gem once the agent backtracks to the second key (panel iv). In Figure 1b, SIPS remains uncertain about all three goals when the first key is acquired (panel ii), but discards the red gem as a possibility when the agent walks past the door (panel iii), and finally converges upon the blue gem when the agent myopically unlocks the first door required to access that gem (panel iv).

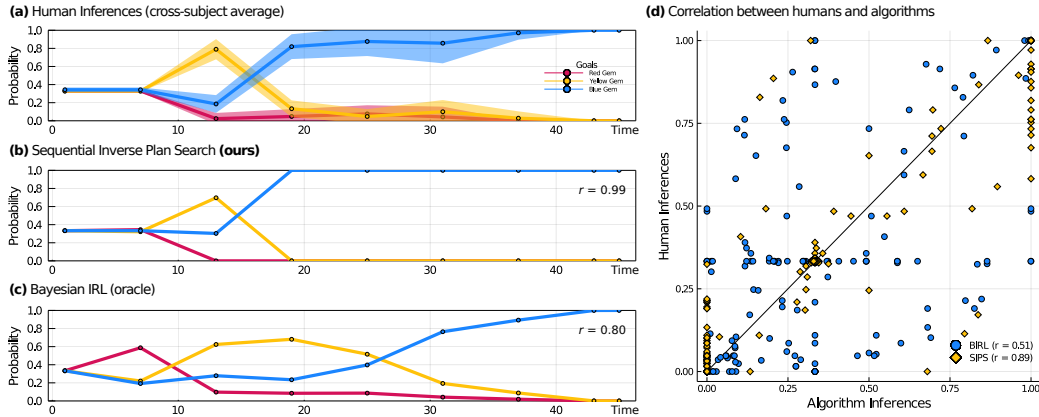

**Figure 4: (a)** Average human goal inferences over time ($\pm 1$ std.) for the sub-optimal trajectory in Figure 1a, compared to **(b)** inferences made by SIPS and **(c)** oracle BIRL. We omit unbiased BIRL because unbiased VI fails to converge for this domain, producing a flat posterior. In **(d)** we show a scatterplot of mean human inferences against algorithm inferences across all trajectories.

In addition to posterior convergence, the inferences made by SIPS display human-like changes at key timepoints. We quantified this human-likeness by collecting human goal inferences on ten trajectories (six sub-optimal or failed) in a pilot study with $N$=8 subjects. Human inferences were collected every six timesteps, and a comparison against SIPS and the oracle BIRL baseline is shown in Figure 4. For the trajectory in in Figure 1a, human inferences (Figure 4a) display extremely similar qualitative trends as SIPS (Figure 4b, $r$=0.99). Oracle BIRL correlates less well (Figure 4c, $r$=0.80), assigning high probability to the yellow gem even after the agent backtracks at $t \geq 18$. This is because Boltzmann action noise assigns significant likelihood to the undoing of past actions. As Figure 4d shows, SIPS also correlates more strongly with mean human inferences across the dataset. Inferences made SIPS (yellow) hew closely to the diagonal, achieving a correlation of $r$=0.89, indicating that the agent model assumed by SIPS is similar to humans' theory-of-mind. In contrast, inferences made by BIRL (blue) are much more diffuse, achieving a correlation of only $r$=0.51.

## 5.4 Accuracy, Speed and Robustness of Inference

To evaluate accuracy and speed, we ran each inference method on a dataset of optimal and non-optimal agent trajectories for each domain, assuming a uniform prior over goals. The optimal trajectories were generated using A* search with an admissible heuristic for each possible goal in the domain. Non-optimal trajectories were generated using the replanning agent model in Figure 3b, with parameters $r$=2, $q$=0.95, $\gamma$=0.1. We found that with matched model parameters, SIPS achieved good performance with 10 particles per goal without the use of rejuvenation moves, so we report those results here. Further experimental details and parameters can be found in the supplement.

We summarize the results of these experiments in Table 1a, with additional results in the supplement. Our method greatly outperforms the unbiased BIRL baseline in both accuracy and speed in three out of four domains, with an average runtime (AC) often several orders of magnitude smaller. This is largely because unbiased VI fails to converge except for the highly restricted Taxi domain. In contrast, SIPS requires far less initial computation, albeit with higher marginal cost due its online generation of partial plans. In fact, it achieves comparable accuracy and speed to the oracle BIRL baseline, sometimes with less computation (e.g. in Doors, Keys & Gems). SIPS also produces higher estimates of the goal posterior $P(g_{\text{true}}|o)$. This is a reflection of the underlying agent model, which assumes randomness at the level of planning instead of acting. As a result, even a few observations can provide substantial evidence that a particular plan and goal was chosen.

Given the specific assumptions made by our agent model, a reasonable question is whether inference is robust to plans generated by other agent models or actual humans. To address this, we also performed a series of robustness experiments for two domains (Table 1b) on data generated by mismatched model parameters $r$, $q$, $\gamma$, mismatched planning heuristics $h$, Boltzmann-rational RL agents, optimal agents, and 5 pilot human subjects (30 trajectories per subject).

| | | Accuracy | | | | | | Runtime | | | |
|---|---|---|---|---|---|---|---|---|---|---|---|
| | | $P(g_{\text{true}}|o)$ | | | Top-1 | | | | | | |
| **Domain** | **Method** | Q1 | Q2 | Q3 | Q1 | Q2 | Q3 | $C_0$ (s) | MC (s) | AC (s) | N |
| Taxi (3 Goals) | SIPS (ours) | **0.44** | **0.50** | 0.62 | **0.53** | **0.56** | 0.67 | 13.0 | 1.80 | 2.55 | **1429** |
| | BIRL (unbiased) | 0.34 | 0.35 | **0.79** | 0.33 | 0.42 | **0.92** | **2.22** | **0.00** | **0.16** | 10000 |
| | BIRL (oracle) | 0.37 | 0.47 | 0.81 | 0.42 | 0.44 | 0.86 | 1.63 | 0.00 | 0.12 | 2500 |
| Doors, Keys & Gems (3 Goals) | SIPS (ours) | **0.37** | **0.51** | **0.61** | **0.74** | **0.74** | **0.74** | **3.30** | 0.70 | **0.86** | **2099** |
| | BIRL (unbiased) | 0.33 | 0.33 | 0.33 | 0.33 | 0.33 | 0.33 | 3326 | **0.12** | 154 | 250000 |
| | BIRL (oracle) | 0.37 | 0.36 | 0.42 | 0.44 | 0.60 | 0.80 | 150 | 0.12 | 7.01 | 10000 |
| Block Words (5 Goals) | SIPS (ours) | **0.47** | **0.83** | **0.90** | **0.78** | **0.84** | **0.91** | 20.8 | 2.46 | **4.15** | **2506** |
| | BIRL (unbiased) | 0.20 | 0.20 | 0.21 | 0.42 | 0.49 | 0.56 | 687 | **0.27** | 63.6 | 250000 |
| | BIRL (oracle) | 0.20 | 0.29 | 0.45 | 0.73 | 0.80 | 0.96 | 22.2 | 0.05 | 2.12 | 10000 |
| Intrusion Detection (20 Goals) | SIPS (ours) | **0.56** | **0.87** | **0.87** | **0.65** | **0.87** | **0.87** | 375 | 6.60 | **28.0** | **13321** |
| | BIRL (unbiased) | 0.05 | 0.05 | 0.05 | 0.05 | 0.05 | 0.05 | 18038 | **0.75** | 1069 | 250000 |
| | BIRL (oracle) | 0.09 | 0.24 | 0.53 | 0.94 | 1.00 | 1.00 | 98 | 0.02 | 6.00 | 10000 |

**(a) Accuracy and runtime of goal inference across domains and inference methods.** We quantify accuracy at the 1st, 2nd and 3rd quartiles (Q1–Q3) of each observed trajectory via the posterior probability of the true goal $P(g_{\text{true}}|o)$, and the fraction of problems where $g_{\text{true}}$ is top-ranked (Top-1). We measure runtime in terms of the start-up cost ($C_0$), marginal cost per timestep (MC), and average cost per timestep (AC) in seconds. We also report the total number of states visited (N) during either search or value iteration as a platform-independent measure. Excluding the oracle baseline, the best metrics are bolded.

| | Persistence ($r$) | | | Persistence ($q$) | | | RL | Optimal |
|---|---|---|---|---|---|---|---|---|
| **Domain** | 1 | 2* | 4 | 0.8 | 0.9 | 0.95* | $\alpha$=50 | |
| Doors, Keys, Gems | 0.60 | 0.73 | 0.73 | 0.53 | 0.60 | 0.73 | 0.58 | 0.80 |
| Block Words | 0.90 | 0.87 | 0.90 | 0.70 | 0.83 | 0.87 | 0.82 | 0.80 |

| | Search Noise. ($\gamma$) | | | Heuristic ($h$) | | | | Humans |
|---|---|---|---|---|---|---|---|---|
| **Domain** | 0.5 | 0.1* | 0.02 | Mh.* | Mz. | GC. | $h_{\text{add}}$* | $n$=5 |
| Doors, Keys, Gems | 0.67 | 0.73 | 0.77 | 0.73 | 0.90 | – | – | 0.79 |
| Block Words | 0.83 | 0.87 | 0.87 | – | – | 0.43 | 0.87 | 0.73 |

**(b) Robustness to model mismatch.** Top-1 accuracy of SIPS at the third time quartile (Q3), evaluated on data generated by mismatched parameters, Boltzmann-rational RL agents, optimal agents, and humans. We ran SIPS assuming $r$=2, $q$=0.95, $T$=10. For Doors, Keys, Gems, we assumed a Manhattan (Mh.) heuristic against a maze distance (Mz.) heuristic. For Block Words, we assumed $h_{\text{add}}$ against the naive goal count (GC.) heuristic. Matched parameters are starred (*).

**Table 1:** Accuracy, runtime, and robustness of inference.

As Table 1b shows, SIPS is relatively robust to data generated by these other models and parameters. Although performance can degrade with mismatch, this is partly due to the difficulty of inference from highly random behavior (e.g. $q$=0.8, $h$=GC.). On the other hand, when mismatched parameters are *more* optimal, performance can *improve* (e.g. $h$=Mz.). Importantly, SIPS also does well on human data, showing robustness even when the planner is unknown. While our boundedly rational agent model cannot possibly capture all aspects of human planning, these experiments suggest that it is serves as a reasonable approximation, similar to our intuitive theories of other people's minds.

## 6 Limitations and Future Work

In this paper, we demonstrated an architecture capable of online inference of goals and plans, even when those plans might fail. However, important limitations remain. First, we considered only finite sets of goals, but the space of goals that humans pursue is easily infinite. Relatedly, we assume that these goals are final, instead of accounting for the hierarchical and instrumental nature of goals and plans. A promising next step would thus be to express hierarchies of goals and plans as probabilistic grammars or programs [43, 44, 45], capturing both the infinitude and structure of the motives we attribute to each other [46, 47]. Second, unlike the domains considered here, the environments we operate in often involve stochastic dynamics and infinite action spaces [48, 49]. A natural extension would be to integrate Monte Carlo Tree Search or sample-based motion planners into our architecture as modeling components [23], potentially parameterized by learned heuristics [50]. With hope, our architecture might then approach the full complexity of problems that we face everyday, whether one is stacking blocks as a kid, finding the right keys for the right doors, or writing a research paper.

## 7 Broader Impact

We embarked upon this research in the belief that, as increasingly powerful autonomous systems become embedded in our society, it may eventually become necessary for them to accurately understand our goals and values, so as to robustly act in our collective interest. Crucially, this will require such systems to understand the ways in which humans routinely fail to achieve our goals, and not take that as evidence that those goals were never desired. Due to our manifold cognitive limitations, gaps emerge between our goals and our intentions, our intentions and our actions, our beliefs and our conclusions, and our ideals and our practices. To the extent that we would like machines to aid us in actualizing the goals and ideals we most value, rather than those we appear to be acting towards, it will be critical for them to understand how, when, and why those gaps emerge. This aspect of the value alignment problem has thus far been under-explored [51]. By performing this research at the intersection of cognitive science and AI, we hope to lay some of the conceptual and technical groundwork that may be necessary to understand our boundedly-rational behavior.

Of course, the ability to infer the goals of others, and to do so online and despite failures, has many more immediate uses, each of them with its own set of benefits and risks. Perhaps the most straightforwardly beneficial are assistive use cases, such as smart user interfaces [52], intelligent personal assistants, and collaborative robots, which may offer to aid a user if that user appears to be pursuing a sub-optimal plan. However, even those use cases come with the risk of reducing human autonomy, and care should be taken so that such applications ensure the autonomy and willing consent of those being aided [53].

More concerning however is the potential for such technology to be abused for manipulative, offensive, or surveillance purposes. While the research presented in this paper is nowhere near the level of integration that would be necessary for active surveillance or manipulation, it is highly likely that mature versions of similar technology will be co-opted for such purposes by governments, militaries, and the security industry [54, 55]. Although detecting and inferring "suspicious intent" may not seem harmful in its own right, these uses need to be considered within the broader context of society, especially the ways in which marginalized peoples are over-policed and incarcerated [56]. Given these risks, we urge future research on this topic to consider seriously the ways in which technology of this sort will most likely be used, by which institutions, and whether those uses will tend to lead to just and beneficial outcomes for society as a whole. The ability to infer and understand the motives of others is a skill that can be wielded to both great benefit and great harm. We ought to use it wisely.

## 8 Code Availability

Code for the architecture and experiments presented in this paper is available at `https://github.com/ztangent/Plinf.jl/tree/neurips-2020-experiments`, as part of the `Plinf.jl` package for Bayesian inverse planning.

## 9 Acknowledgements

This work was funded in part by the DARPA Machine Common Sense program (Award ID: 030523-00001); philanthropic gifts from the Aphorism Foundation and from the Siegel Family Foundation; and financial support from the MIT-IBM Watson AI Lab and the Intel Probabilistic Computing Center. Tom Silver is supported by an NSF Graduate Research Fellowship.

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
