[Supplementary Material]

# Supplemental Material

## Online Bayesian Goal Inference for Boundedly-Rational Planning Agents

**Tan Zhi-Xuan,   Jordyn L. Mann,   Tom Silver**
**Joshua B. Tenenbaum,      Vikash K. Mansinghka**
Massachusetts Institute of Technology
{xuan,jordynm,tslvr,jbt,vkm}@mit.edu

## A   Experimental Details

Below we provide experimental details for each of the inference methods described in the main text. We have also performed additional experiments using a baseline adapted from the plan recognition as planning (PRP) literature [1], which we include below as a useful offline benchmark.

### A.1   Sequential Inverse Plan Search

We conducted experiments using two main variants of Sequential Inverse Plan Search (SIPS), the first using data-driven rejuvenation, as described in the main text, and the second without. Rejuvenation is necessary for the results shown in Figure 1 of the main text, and for highly sub-optimal and failed plans more generally. However, rejuvenation is also hard to tune, and can increase runtime due to the need to replan. We thus report results without rejuvenation in our quantitative experiments.

Parameters for qualitative experiments are given in each of the corresponding figures in section B.1. For the quantitative experiments, we used SIPS with 10 particles per possible goal (e.g., 50 particles for the Block Words domain), with a resampling threshold of $c = 1/4$. For the underlying agent model, we assumed search noise of $\gamma = 0.1$ and persistence parameters of $r = 2$ and $q = 0.95$ (giving an average search budget of 38 nodes). We varied the search heuristic $h$ to suit the type of domain: For the gridworld-based domains (Taxi; Doors, Keys & Gems), we used a Manhattan distance heuristic to the goal. For the other domains (Block Words; Intrusion Detection), we used the $h_{\mathrm{add}}$ heuristic introduced by the HSP algorithm [2] as a generalized relaxed-distance heuristic.

SIPS also requires the specification of an observation model $P(o|s)$, in order to score the likelihood of a hypothesized state trajectory $\hat{s}_1, ..., \hat{s}_t$ given the observed states $o_1, ..., o_t$. We defined this observation model by adding zero-mean Gaussian noise with $\sigma = 0.25$ for each numeric variable in the state (e.g., the agent's position in a gridworld), and Bernoulli corruption noise with $p = 0.05$ for each Boolean variable in the state (e.g. whether block A is on top of block B).

All SIPS experiments were performed using `Plinf.jl`, a Julia implementation of our modeling and inference architecture that integrates the Gen probabilistic programming system with `PDDL.jl`, a Julia interpreter for the Planning Domain Definition Language [3]. Experiments were run on a 1.9 GHz Intel Core i7 processor with 16 GB RAM.

### A.2   Bayesian Inverse Reinforcement Learning

Bayesian Inverse Reinforcement Learning (BIRL) requires computing an approximate value function $Q(s, a)$ offline and a posterior over goals online using the likelihood $P(a \mid s, g) = \frac{1}{Z} e^{\alpha \cdot Q(s,a)}$, where $Z$ is the partition function and $\alpha$ is an optimality parameter. For the quantitative experiments, we used $\alpha = 1$ which we found to perform well in preliminary trials. For qualitative comparisons, however, we used $\alpha = 5$, as this choice produced results in range more similar to human inferences. To approximate the value function, we used value iteration (VI) with a discount factor of $0.9$.

As discussed in the main text, several of the domains considered in this work have state spaces that are too large to enumerate, making standard VI intractable. We therefore used asynchronous VI, sampling states instead of fully enumerating them, for 250,000 iterations for the unbiased baseline (BIRL-U). Preliminary experiments suggested that running for up to 1,000,000 iterations did not appreciably improve results. Taxi, which has a far smaller state space than the other domains, was run with 10,000 iterations, which was consistently sufficient for convergence. For the oracle baseline (BIRL-O), 2500 iterations were sufficient to reach convergence for the Taxi domain, and 10,000 iterations for the other domains.

All BIRL experiments were written in Python and run on a 2.9 GHz Intel Core i9 processor with 32 GB RAM. We made use of the PDDLGym library [4] for instantiating the PDDL planning problems as OpenAI Gym environments. To perform asynchronous VI efficiently, we implemented state samplers and valid action generators for each domain. The unbiased version of BIRL (BIRL-U) uses these state samplers to sample states within asynchronous VI. For the oracle baseline (BIRL-O), which has access to the test-time trajectories, we instead sampled one state uniformly at random from the states visited across all test-time trajectories.

### A.3 Plan Recognition as Planning

We adapted the plan recognition as planning (PRP) approach described in [1] as an offline benchmark that achieves high accuracy at the cost of substantially more runtime (up to 30 times) than SIPS. In the PRP approach, we use a heuristic approximation to the likelihood of a plan $p$ given a goal $g$:

$$P(p|g) \propto e^{-\beta(|p|-|p_*^g|)} \tag{1}$$

where $p_*^g$ is an optimal plan to the goal $g$, $|p|$ denotes the length of the plan $p$, and $\beta$ is a noise parameter. This likelihood function model agent rationality by placing exponentially less probability on costlier plans, where larger values of $\beta$ correspond to more optimality.

In order to perform inference using this likelihood model, we first compute the optimal plan $p_*^g$ for each possible goal $g$ in a domain. At each timestep $t$, we then construct a plan $p_t^g$ to each goal $g$ consistent with the observations so far, by computing an optimal partial plan $p_t^+$ from the current observed state $o_t$ to $g$, and then concatenating it with the initial sequence of actions $p_t^- := a_1, ..., a_{t-1}$ taken by the agent, giving $p_t^g = [p_t^-, p_t^+]$. Under the additional approximation that $p_t^g$ is the only plan consistent with the observation sequence $o_1, ..., o_t$, we can then compute the goal posterior as

$$P(g|o_1, ..., o_t) \simeq \frac{e^{-\beta(|p_t^g|-|p_*^g|)}}{\sum_{g' \in \mathcal{G}} e^{-\beta(|p_t^{g'}|-|p_*^{g'}|)}} \tag{2}$$

The main limitation of this approach is that it requires computation of an optimal partial plan $p_t^+$ for every goal $g$ at every timestep $t$, which scales poorly with the number of goals and timesteps per trajectory, especially when the observed trajectory leads the agent further and further away from most of the goals under consideration. This is contrast to SIPS, which performs incremental computation by extending partial plans from previous timesteps. In addition, due to the assumption that there always exists a plan from the current observed state $o_t$ to every goal $g$, the PRP approach is unable to account for irreversible failures. This is shown in our qualitative comparisons.

Nonetheless, because PRP still achieves high accuracy on many sub-optimal trajectories (at the expense of considerably more computation, especially on domains with many goals), we include it here as a benchmark for accuracy. All PRP experiments were performed on the same machine as the SIPS experiments, using the implementation of A* search provided by `Plinf.jl`.

## B  Additional Results

### B.1  Qualitative Comparisons for Sub-Optimal & Failed Plans

Here we present detailed qualitative comparisons of the goal inferences made for sub-optimal and failed plans in the Doors, Keys & Gems domain. Figures S1 and S2 show the inferences made for two sub-optimal trajectories, while Figures S3 and S4 show the inferences made for two trajectories with irreversible failures. We omit the unbiased Bayesian IRL baseline (BIRL-U), because it is unable to solve the underlying Markov Decision Process in any of these examples, leading to a uniform posterior over goals over the entire trajectory.

**Figure S1:** Goal inferences made by SIPS, BIRL-O, and PRP for the sub-optimal trajectory shown in Figure 1(a) of the main text. Predicted future trajectories in panels (i)–(iv) are made by SIPS. For SIPS, we used 30 particles per goal, search noise $\gamma = 0.1$, persistence parameters $r = 2$, $q = 0.95$, and a Manhattan distance heuristic to the goal. Rejuvenation moves were used, with a goal rejuvenation probability of $p_g = 0.25$. For BIRL-O, we used $\alpha = 5$. For PRP, we used $\beta = 1$.

**Figure S2:** Goal inferences made by SIPS, BIRL-O, and PRP for another sub-optimal trajectory. Predicted future trajectories in panels (i)–(iv) are made by SIPS. For SIPS, we used 30 particles per goal, search noise $\gamma = 0.1$, persistence parameters $r = 2$, $q = 0.95$, and a Manhattan distance heuristic to the goal. Rejuvenation moves were used, with a goal rejuvenation probability of $p_g = 0.25$. For BIRL-O, we used $\alpha = 5$. For PRP, we used $\beta = 1$.

### B.1.1 Sub-Optimal Plans

Figure S1 shows how the inferences produced by SIPS are more human-like, compared to the BIRL and PRP baselines. In particular, SIPS adjusts its inferences in a human-like manner, initially remaining uncertain between the 3 gems (panel i), placing more posterior mass on the yellow gem when the agent acquires the first key (panel ii), increasing that posterior mass when agent appears to ignore the second key and unlock the first door (panel iii), but then switching to the blue gem once the agent backtracks towards the second key (panel iv).

While the inferences produced by BIRL display similar trends, they are much more gradual, because BIRL assumes noise at the level of acting instead of planning. In addition, the agent model underlying BIRL leads to strange artifacts, such as the rise in probability of the red gem when $t < 9$. This is because Boltzmann action noise places lower probability $P(a|g)$ on an action $a$ that leads to a goal $g$ which is further away, due to the value function $V_g$ associated with that goal $g$ being smaller due to time discounting. As a result, when $t < 9$, BIRL computes that $P(\texttt{right}|\texttt{red}) > P(\texttt{right}|\texttt{yellow})$ and $P(\texttt{right}|\texttt{blue})$, leading to the red gem being inferred as the most likely goal.

Finally, PRP exhibits both over-confidence in the yellow gem and slow recovery towards the blue gem. This is due to the assumption that the likelihood of a plan $p$ to some goal $g$ is exponentially decreasing in its cost difference from the optimal plan $p_*^g$. Between $t = 10$ and $t = 20$, all plans consistent with the observations to the blue gem are considerably longer than the optimal plan $p_*^{\texttt{blue}}$. As a result, PRP gives very low probability to the blue gem. This effect continues for many timesteps after the agent starts to backtrack ($t = 17$ to $t = 24$), indicating that the PRP modeling assumptions are inadequate for plans with substantial backtracking.

Similar dynamics can be observed for the trajectory in Figure S2. The BIRL baseline performs especially poorly, placing high probability on the yellow gem even when the agent backtracks to collect the second key ($t = 19$ to $t = 22$). This again is due to the assumption of action noise instead of planning noise, making it much more likely under the BIRL model that an agent would randomly walk back towards the second key. The PRP baseline exhibits the same issues with over-confidence and slow recovery described earlier, placing so little posterior mass on the blue gem from $t = 17$ to $t = 20$ that it even considers the red gem to be more likely. In contrast, our method, SIPS, immediately converges to the blue gem once backtracking occurs at $t = 20$.

### B.1.2 Failed Plans

The differences between SIPS and the baseline methods are even more striking for trajectories with irreversible failures. As shown in Figure S3, SIPS accurately infers that the blue gem is the most likely goal when the agent ignores the two keys at the bottom, instead turning towards the first door guarding the blue gem at $t = 19$. This inference also remains stable after $t = 21$, when the agent irreversibly uses up its key to unlock that door. SIPS is capable of such inferences because the search for partial plans is biased towards promising intermediate states. Since the underlying agent model assumes a relaxed distance heuristic that considers states closer to the blue gem as promising, the model is likely to produce partial plans that lead spatially toward the blue gem, even if those plans myopically use up the agent's only key.

In contrast, both BIRL and PRP fail to infer that the blue gem is the goal. BIRL initially places increasing probability on the red gem, due to Boltzmann action noise favoring goals which take less time to reach. While this probability decreases slightly as the agent detours from the optimal plan to the red gem, it remains the highest probability goal even after the agent uses up its key at $t = 21$. The posterior over goals stops changing after that, because there are no longer any any possible paths to a goal. PRP exhibits a different failure mode. While it does not suffer from the artifacts due to Boltzmann action noise, it completely fails to account for the possibility that an agent might make a failed plan. As a result, the probability of the blue gem does not increase even after the agent turns towards it at $t = 19$. Furthermore, once failure occurs at $t = 21$, PRP ends up defaulting to a uniform distribution over the three gems, even though it had previously eliminated the red gem as a possibility.

The inferences in Figure S4 display similar trends. Once again, SIPS accurately infers that the blue gem is the goal, even slightly in advance of failure (panel iii). In contrast, BIRL wrongly infers that the red gem is the most likely, while PRP erroneously defaults to inferring upon failure that the only remaining acquirable gem (yellow) is the goal.

**Figure S3:** Goal inferences made by SIPS, BIRL-O, and PRP for the failed trajectory shown in Figure 1(b) of the main text. Predicted future trajectories in panels (i)–(iv) are made by SIPS. For SIPS, we used 30 particles per goal, search noise $\gamma = 0.1$, persistence parameters $r = 2$, $q = 0.95$, and a maze-distance heuristic (i.e. distance to the goal, ignoring doors). Rejuvenation moves were used with $p_g = 0.25$. For BIRL-O, we used $\alpha = 5$. For PRP, we used $\beta = 1$.

**Figure S4:** Goal inferences made by SIPS, BIRL-O, and PRP for another failed trajectory. Predicted future trajectories in panels (i)–(iv) are made by SIPS. For SIPS, we used 30 particles per goal, search noise $\gamma = 0.1$, persistence parameters $r = 2$, $q = 0.95$, and a Manhattan distance heuristic to the goal. Rejuvenation moves were used, with a goal rejuvenation probability of $p_g = 0.25$. For BIRL-O, we used $\alpha = 5$. For PRP, we used $\beta = 1$.

## B.2 Accuracy & Speed

Here we present quantitative comparisons of the accuracy and speed of each inference method. Tables S1 and S2 show the accuracy results for the optimal and sub-optimal datasets respectively. $P(g_{\text{true}}|o)$ represents the posterior probability of the true goal, while Top-1 represents the fraction of problems where $g_{\text{true}}$ is top-ranked. Accuracy metrics are reported at the first (Q1), second (Q2), and third (Q3) quartiles of each observed trajectory. The corresponding standard deviations (taken across the dataset) are shown to the right of each accuracy mean.

Tables S3 and S4 show the runtime results for the optimal and sub-optimal datasets respectively. Runtime is reported in terms of the start-up cost ($C_0$), marginal cost per timestep (MC), and average cost per timestep (AC), all measured in seconds. The corresponding standard deviations are shown to the right of each runtime mean. The total number (N) of states visited (during either plan search or value iteration) are also reported as a platform-independent cost metric.

| Domain | Method | $P(g_{\text{true}}\|o)$ Q1 | Q2 | Q3 | Top-1 Q1 | Q2 | Q3 |
|---|---|---|---|---|---|---|---|
| Taxi (3 Goals) | SIPS | 0.45 ±0.26 | 0.48 ±0.27 | 0.64 ±0.32 | 0.67 ±0.49 | 0.67 ±0.49 | 0.67 ±0.49 |
| | BIRL-U | 0.33 ±0.06 | 0.38 ±0.17 | 0.79 ±0.22 | 0.33 ±0.47 | 0.42 ±0.49 | 0.92 ±0.28 |
| | BIRL-O | 0.41 ±0.33 | 0.44 ±0.40 | 0.82 ±0.23 | 0.50 ±0.50 | 0.42 ±0.49 | 1.00 ±0.00 |
| | PRP | 0.33 ±0.00 | 0.36 ±0.06 | 0.44 ±0.08 | 0.33 ±0.00 | 1.00 ±0.00 | 1.00 ±0.00 |
| Doors, Keys & Gems (3 Goals) | SIPS | 0.39 ±0.18 | 0.51 ±0.32 | 0.70 ±0.35 | 0.73 ±0.46 | 0.73 ±0.46 | 0.80 ±0.41 |
| | BIRL-U | 0.33 ±0.00 | 0.33 ±0.00 | 0.33 ±0.00 | 0.33 ±0.00 | 0.33 ±0.00 | 0.33 ±0.00 |
| | BIRL-O | 0.41 ±0.33 | 0.37 ±0.06 | 0.41 ±0.08 | 0.50 ±0.50 | 0.67 ±0.47 | 0.87 ±0.34 |
| | PRP | 0.40 ±0.17 | 0.62 ±0.30 | 0.81 ±0.26 | 1.00 ±0.00 | 1.00 ±0.00 | 1.00 ±0.00 |
| Block Words (5 Goals) | SIPS | 0.38 ±0.27 | 0.71 ±0.41 | 0.78 ±0.41 | 0.73 ±0.46 | 0.73 ±0.46 | 0.80 ±0.41 |
| | BIRL-U | 0.20 ±0.03 | 0.21 ±0.05 | 0.23 ±0.10 | 0.53 ±0.50 | 0.53 ±0.50 | 0.60 ±0.49 |
| | BIRL-O | 0.22 ±0.01 | 0.30 ±0.03 | 0.46 ±0.06 | 0.73 ±0.44 | 0.87 ±0.34 | 1.00 ±0.00 |
| | PRP | 0.38 ±0.18 | 0.78 ±0.28 | 0.91 ±0.18 | 0.93 ±0.26 | 0.93 ±0.26 | 1.00 ±0.00 |
| Intrusion Detection (20 Goals) | SIPS | 0.65 ±0.38 | 1.00 ±0.00 | 1.00 ±0.00 | 0.80 ±0.41 | 1.00 ±0.00 | 1.00 ±0.00 |
| | BIRL-U | 0.05 ±0.00 | 0.05 ±0.00 | 0.05 ±0.00 | 0.05 ±0.00 | 0.05 ±0.00 | 0.05 ±0.00 |
| | BIRL-O | 0.10 ±0.01 | 0.25 ±0.02 | 0.55 ±0.03 | 1.00 ±0.00 | 1.00 ±0.00 | 1.00 ±0.00 |
| | PRP | 0.35 ±0.13 | 0.96 ±0.06 | 0.99 ±0.01 | 1.00 ±0.00 | 1.00 ±0.00 | 1.00 ±0.00 |

**Table S1:** Inference accuracy on the dataset of optimal trajectories.

| Domain | Method | $P(g_{\text{true}}\|o)$ Q1 | Q2 | Q3 | Top-1 Q1 | Q2 | Q3 |
|---|---|---|---|---|---|---|---|
| Taxi (3 Goals) | SIPS | 0.43 ±0.32 | 0.51 ±0.38 | 0.62 ±0.42 | 0.46 ±0.51 | 0.50 ±0.51 | 0.67 ±0.48 |
| | BIRL-U | 0.34 ±0.06 | 0.33 ±0.00 | 0.79 ±0.23 | 0.33 ±0.47 | 0.42 ±0.49 | 0.92 ±0.28 |
| | BIRL-O | 0.35 ±0.29 | 0.48 ±0.32 | 0.81 ±0.32 | 0.38 ±0.48 | 0.46 ±0.50 | 0.79 ±0.41 |
| | PRP | 0.33 ±0.00 | 0.35 ±0.06 | 0.53 ±0.23 | 0.33 ±0.00 | 1.00 ±0.00 | 1.00 ±0.00 |
| Doors, Keys & Gems (3 Goals) | SIPS | 0.35 ±0.07 | 0.51 ±0.32 | 0.54 ±0.37 | 0.75 ±0.44 | 0.75 ±0.44 | 0.70 ±0.47 |
| | BIRL-U | 0.33 ±0.00 | 0.33 ±0.00 | 0.33 ±0.00 | 0.33 ±0.00 | 0.33 ±0.00 | 0.33 ±0.00 |
| | BIRL-O | 0.34 ±0.02 | 0.36 ±0.04 | 0.43 ±0.07 | 0.4 ±0.49 | 0.55 ±0.50 | 0.75 ±0.43 |
| | PRP | 0.35 ±0.17 | 0.38 ±0.32 | 0.64 ±0.40 | 0.90 ±0.31 | 0.70 ±0.47 | 0.83 ±0.38 |
| Block Words (5 Goals) | SIPS | 0.52 ±0.33 | 0.89 ±0.28 | 0.96 ±0.18 | 0.80 ±0.41 | 0.90 ±0.31 | 0.97 ±0.18 |
| | BIRL-U | 0.19 ±0.03 | 0.19 ±0.03 | 0.19 ±0.04 | 0.37 ±0.48 | 0.47 ±0.50 | 0.53 ±0.50 |
| | BIRL-O | 0.19 ±0.03 | 0.29 ±0.06 | 0.45 ±0.09 | 0.73 ±0.44 | 0.77 ±0.42 | 0.93 ±0.25 |
| | PRP | 0.36 ±0.18 | 0.77 ±0.24 | 0.91 ±0.17 | 1.00 ±0.00 | 1.00 ±0.00 | 1.00 ±0.00 |
| Intrusion Detection (20 Goals) | SIPS | 0.52 ±0.43 | 0.80 ±0.41 | 0.80 ±0.41 | 0.58 ±0.50 | 0.80 ±0.41 | 0.80 ±0.41 |
| | BIRL-U | 0.05 ±0.00 | 0.05 ±0.00 | 0.05 ±0.00 | 0.05 ±0.00 | 0.05 ±0.00 | 0.05 ±0.00 |
| | BIRL-O | 0.09 ±0.01 | 0.23 ±0.04 | 0.52 ±0.07 | 0.92 ±0.22 | 1.00 ±0.00 | 1.00 ±0.00 |
| | PRP | 0.42 ±0.01 | 0.99 ±0.003 | 1.00 ±0.00 | 1.00 ±0.00 | 1.00 ±0.00 | 1.00 ±0.00 |

**Table S2:** Inference accuracy on the dataset of suboptimal trajectories.

In terms of accuracy alone, it can be seen that the PRP baseline generally achieves the highest metrics, with SIPS and BIRL-O performing comparably, and with BIRL-U completely incapable of making accurate inferences except in the Taxi domain. As demonstrated by the qualitative comparisons however, these metrics alone maybe misleading, failing to show how inferences of each method really evolve over time. In particular, while the PRP baseline is routinely able to achieve the highest Top-1 accuracy, this may not correspond to a suitably calibrated posterior over goals, nor might it capture the sharp human-like changes over time that SIPS appears to display. It should also be noted that most of the domains considered do not allow for irreversible failures. As such, the distinctive capability of SIPS to infer goals despite failed plans is not captured by the results in Table S2.

| Domain | Method | C0 (s) | | MC (s) | | AC (s) | | N | |
|---|---|---|---|---|---|---|---|---|---|
| | | | | | | | | **Runtime** | |
| Taxi (3 Goals) | SIPS | 14.7 | ±6.73 | 2.19 | ±0.95 | 3.08 | ±1.24 | 1220 | ±405 |
| | BIRL-U | 2.22 | ±0.06 | 0.002 | ±0.0007 | 0.17 | ±0.03 | 10000 | ±0 |
| | BIRL-O | 0.56 | ±0.02 | 0.002 | ±0.0006 | 0.04 | ±0.01 | 2500 | ±0 |
| | PRP | 13.2 | ±2.19 | 6.21 | ±1.52 | 6.73 | ±1.50 | 6830 | ±2090 |
| Doors, Keys & Gems (3 Goals) | SIPS | 3.17 | ±1.10 | 0.72 | ±0.21 | 0.84 | ±0.25 | 2100 | ±1140 |
| | BIRL-U | 3280 | ±173 | 0.13 | ±0.14 | 181 | ±184 | 250000 | ±0 |
| | BIRL-O | 142 | ±13.0 | 0.13 | ±0.14 | 8.00 | ±8.24 | 10000 | ±0 |
| | PRP | 5.32 | ±2.21 | 3.12 | ±1.58 | 3.24 | ±1.67 | 5970 | ±3350 |
| Block Words (5 Goals) | SIPS | 21.1 | ±4.84 | 1.67 | ±0.61 | 3.62 | ±0.85 | 2380 | ±1110 |
| | BIRL-U | 687 | ±273 | 0.15 | ±0.05 | 69.5 | ±31.2 | 250000 | ±0 |
| | BIRL-O | 19.5 | ±0.59 | 0.12 | ±0.03 | 2.11 | ±0.51 | 10000 | ±0 |
| | PRP | 25.6 | ±11.3 | 26.5 | ±7.90 | 26.3 | ±7.50 | 3980 | ±1410 |
| Intrusion Detection (20 Goals) | SIPS | 325 | ±24.9 | 12.0 | ±1.40 | 30.0 | ±3.00 | 14100 | ±343 |
| | BIRL-U | 18000 | ±2050 | 0.01 | ±0.07 | 1130 | ±230 | 250000 | ±0 |
| | BIRL-O | 100 | ±11.7 | 0.02 | ±0.00 | 5.80 | ±0.86 | 10000 | ±0 |
| | PRP | 246 | ±5.12 | 381 | ±108 | 374 | ±102 | 75700 | ±20800 |

**Table S3:** Inference runtime on the dataset of optimal trajectories.

| Domain | Method | C0 (s) | | MC (s) | | AC (s) | | N | |
|---|---|---|---|---|---|---|---|---|---|
| | | | | | | | | **Runtime** | |
| Taxi (3 Goals) | SIPS | 12.2 | ±7.75 | 1.61 | ±0.74 | 2.29 | ±1.05 | 1530 | ±1110 |
| | BIRL-U | 2.22 | ±0.06 | 0.003 | ±0.0004 | 0.16 | ±0.04 | 10000 | ±0.00 |
| | BIRL-O | 2.17 | ±0.05 | 0.002 | ±0.0003 | 0.15 | ±0.04 | 2500 | ±0.00 |
| | PRP | 13.3 | ±3.26 | 7.33 | ±2.61 | 7.74 | ±2.56 | 8840 | ±5800 |
| Doors, Keys & Gems (3 Goals) | SIPS | 3.40 | ±1.18 | 0.69 | ±0.24 | 0.87 | ±0.31 | 2100 | ±1140 |
| | BIRL-U | 3360 | ±66.0 | 0.11 | ±0.06 | 133 | ±68.7 | 250000 | ±0.00 |
| | BIRL-O | 155 | ±3.31 | 0.11 | ±0.06 | 6.27 | ±3.31 | 10000 | ±0.00 |
| | PRP | 4.65 | ±1.58 | 3.04 | ±1.56 | 3.11 | ±1.56 | 6150 | ±3680 |
| Block Words (5 Goals) | SIPS | 20.6 | ±5.79 | 2.86 | ±1.12 | 4.41 | ±1.77 | 2570 | ±810 |
| | BIRL-U | 687 | ±273 | 0.33 | ±0.13 | 60.6 | ±34.0 | 250000 | ±0.00 |
| | BIRL-O | 23.5 | ±1.76 | 0.01 | ±0.001 | 2.12 | ±0.86 | 10000 | ±0.00 |
| | PRP | 40.5 | ±22.7 | 38.9 | ±16.1 | 38.9 | ±15.7 | 5660 | ±4860 |
| Intrusion Detection (20 Goals) | SIPS | 400 | ±29.7 | 3.90 | ±1.04 | 26.6 | ±2.06 | 12900 | ±3020 |
| | BIRL-U | 18000 | ±2050 | 1.12 | ±3.83 | 1040 | ±163 | 250000 | ±0.00 |
| | BIRL-O | 96.9 | ±10.4 | 0.02 | ±0.002 | 5.60 | ±0.77 | 10000 | ±0.00 |
| | PRP | 281 | ±2.48 | 332 | ±25.8 | 330 | ±24.7 | 51900 | ±960 |

**Table S4:** Inference runtime on the dataset of suboptimal trajectories.

Once runtime is taken into account, it becomes clear that SIPS achieves the best balance between speed and accuracy due to its use of incremental computation. In contrast, BIRL-U requires orders of magnitude more initial computation while still failing to produce meaningful inferences, while PRP requires up to 30 times more computation per timestep. This is especially apparent on the Intrusion Detection domain, which has a large number of goals, requiring PRP to compute a large number of optimal plans at each timestep. Even the BIRL-O baseline, which assumes oracular access to the dataset of observed trajectories during value iteration, is slower than SIPS on the Doors, Keys & Gems domain in terms of average runtime. Overall, these results imply that SIPS is the only method suitable for online usage on the full range of domains we consider.

## B.3 Robustness to Parameter Mismatch

Tables S5 and S6 present additional results for the robustness experiments described in the main text, showing how different settings of model parameters fare against each other. Each column corresponds to a parameter value assumed by SIPS, and each row corresponds to the true parameter for the boundedly rational agent model used to generate the data. Within each sub-table, unspecified parameters default to $\gamma = 0.1$, $r = 2$, $q = 0.95$, $h =$ Manhattan (for Doors, Keys, Gems) and $h = h_{\mathrm{add}}$ (for Block Words).

It can be seen that SIPS fares reasonably well against mismatched parameters, with degradation partly driven by mismatch itself, but also partly by increased randomness when the data-generating parameters lead to less optimal agent behavior. The effect of noisy behavior is especially apparent in Table S6(d): data generated by agents using the highly uninformative goal count heuristic (which simply the counts the number of goal predicates yet to be satisfied as a distance metric) is highly random. This results in very poor inferences (Top-1 at Q3 = 0.37), even when SIPS correctly assumes the same heuristic. Nonetheless, mismatched heuristics do lead to poorer performance, raising the open question of whether observers need good models of others' planning heuristics in order to accurately infer their goals.

**(a) Persistence ($r$)**

| True \ Assumed $r$ | 1 | 2 | 4 |
|---|---|---|---|
| 1 | 0.80 | 0.60 | 0.70 |
| 2 | 0.73 | 0.73 | 0.77 |
| 4 | 0.77 | 0.73 | 0.87 |

**(b) Persistence ($q$)**

| True \ Assumed $q$ | 0.80 | 0.90 | 0.95 |
|---|---|---|---|
| 0.80 | 0.73 | 0.67 | 0.53 |
| 0.90 | 0.77 | 0.63 | 0.60 |
| 0.95 | 0.77 | 0.60 | 0.73 |

**(c) Search noise ($\gamma$)**

| True \ Assumed $\gamma$ | 0.02 | 0.10 | 0.50 |
|---|---|---|---|
| 0.02 | 0.90 | 0.77 | 0.83 |
| 0.10 | 0.77 | 0.73 | 0.73 |
| 0.50 | 0.73 | 0.67 | 0.77 |

**(d) Heuristic ($h$)**

| True \ Assumed $h$ | Mh. | Mz. |
|---|---|---|
| Mh. | 0.83 | 0.77 |
| Mz. | 0.80 | 0.90 |

**Table S5:** Robustness to parameter mismatch for the Doors, Keys, Gems domain. The metric shown is the top-1 accuracy of SIPS at the third time quartile (Q3). $h$=Mh. refers to Manhattan distance, while $h$=Mz. refers to maze distance.

**(a) Persistence ($r$)**

| True \ Assumed $r$ | 1 | 2 | 4 |
|---|---|---|---|
| 1 | 0.80 | 0.90 | 0.77 |
| 2 | 0.83 | 0.80 | 0.90 |
| 4 | 0.87 | 0.90 | 0.93 |

**(b) Persistence ($q$)**

| True \ Assumed $q$ | 0.80 | 0.90 | 0.95 |
|---|---|---|---|
| 0.80 | 0.80 | 0.83 | 0.70 |
| 0.90 | 0.70 | 0.80 | 0.83 |
| 0.95 | 0.83 | 0.80 | 0.87 |

**(c) Search noise ($\gamma$)**

| True \ Assumed $\gamma$ | 0.02 | 0.10 | 0.50 |
|---|---|---|---|
| 0.02 | 0.80 | 0.87 | 0.87 |
| 0.10 | 0.83 | 0.87 | 0.83 |
| 0.50 | 0.90 | 0.83 | 0.87 |

**(d) Heuristic ($h$)**

| True \ Assumed $h$ | GC | $h_{\mathrm{add}}$ |
|---|---|---|
| GC. | 0.37 | 0.43 |
| $h_{\mathrm{add}}$ | 0.33 | 0.77 |

**Table S6:** Robustness to parameter mismatch for the Blocks World domain. The metric shown is the top-1 accuracy of SIPS at the third time quartile (Q3). $h$=GC. refers to the goal count heuristic, while $h = h_{\mathrm{add}}$ refers to the additive delete-relaxation heuristic.

# C  Human Studies

As described in the main text, we conducted two sets of pilot studies with human subjects, the first to measure human goal inferences for comparison, and the second to collect human-generated plans for robustness experiments. These studies were approved under MIT's IRB (COUHES no.: 0812003014).

## C.1  Human Inferences

Data was collected from $N$=5 pilot subjects in the MIT population. Each subject was given access to a web interface that would present trajectories of an agent in the Doors, Keys & Gems domain, and that would ask for goal inference judgements at every 6th timestep, as well as the first and last timestep. Subjects could select which gem they believed to be the most likely goal of the agent, and then were allowed to adjust sliders indicating how likely the other goals were in comparison. These relative probability ratings were normalized, and recorded. Excerpts from this interface are shown in Figure S5. Subjects were shown a series of 10 trajectories, out of which 4 were optimal trajectories, and 6 exhibited notable suboptimality or failure.

**Figure S5:** Web interface for collecting human goal inferences. Each panel shows one step in a sequence of judgement points presented to a participant.

## C.2 Human Plans

Data was collected from $N$=5 pilot subjects in the MIT population. Each subject received a experimental script to run, which collected data for both the Doors-Keys-Gems domain and the Blocks World domain. For the Blocks World domain, data was collected for all combinations of 3 problems with 5 possible goals, and for the Doors-Keys-Gems domain, data was collected for all combinations of 5 problems with 3 possible goals.

For each pair consisting of a problem and a goal, the subjects were presented with a visualization of the initial state and a textual description of their goal. The subjects were then presented with a list of keys corresponding to the actions available from the current state, and prompted to press the key corresponding to their selected action. Once the subjects entered their action of choice, the visualization would update to show the state after the action had occurred. The subjects would then be prompted again for an action. This process repeated until the given goal was achieved, or the subject terminated that task (e.g. if the goal was no longer achievable). Once the goal was achieved for a given problem and goal pair, the sequence of actions was recorded.