[Reviews · NeurIPS 2020]

Review 1

Summary and Contributions: The paper presents a framework for inferring goals from online from acting agents, while relaxing assumption that agents perform optionally or approximately optionally and instead assuming that agents exhibit bounded rationality.

Strengths: The paper presents a novel solution to a well-known and actively explored problem, by including the plan itself into the state and modelling plan updating in the Bayesian framework.

Weaknesses: The model of bounded rationality (3.2) in the paper hasn't been quite convincing for me. I am also not sure how important it is for this method to rely on a probabilistic programming framework.

Correctness: Apart from my concerns about section 3.2, there are no significant theoretical results (theorems etc.) proven in the paper, and whatever formal derivations are presented look correct. In the empirical evaluation though, the ability of a system to handle failed and suboptimal plans is tested on hand-crafted cases. I find this not particularly convincing and would prefer a model of an agent prone of suboptimal or failed plans to be used for evaluating the approach.

Clarity: The paper is well written and easy enough to follow.

Relation to Prior Work: The related prior work is covered well.

Reproducibility: Yes

Additional Feedback: Update based on authors' feedback: the authors addressed my concern well enough. I updated my review based on the authors' feedback. Section 3.2: negative-binomial distribution is used to model the persistance of a planner. I do not understand how r (the maximum number of failures (or successes as more often defined)) is related to modeling of a planning attempt. What are the failures (or successes) which occur during planning so that the planner stops after r of them? In Bayesian modelling, it is usually not enough to choose a distribution of an intuitively suitable shape -- at least some justification of the underlying phenomenon is highly desirable. To model limited planning ability of the agent, A* is modified (lines 124--126) so that a randomly selected state around the top state rather than the top state is expanded. I couldn't find a reference to this modification of A* in the literature, and my brief analysis makes me think that A* does not find suboptimal plans faster (as one would expect from a boundedly rational agent) but rather makes the search unsound (it may enter infinite loops and fail). There are boundedly rational A* variants in the literature, both deterministic and stochastic, and they should be used instead. I am also wondering about the importance of reliance on a probabilistic programming framework for this research. I read the source code provided in the supplementary material, and it seems that while some parts of the code look nicer, superfifically, the inference algorithm is implemented essentially from scratch, and the particle weight computation can be implemented without Gen's macros. I would prefer if the paper discussed the need for a probabilistic programming framework in more detail.


Review 2

Summary and Contributions: The paper looks at the problem of online recognition of goals of agents that rely on resource bounded reasoning to come up with plans. The methods discussed in the paper primarily relies on a bayesian formulation for goal recognition and proposes a sequential monte-carlo method to estimate the goal posterior. The approximate possible human planning process using a budget limited noisy planning over PDDL representations of the task. The methods are then evaluated on four domains and they also present some qualitative examples of goal recognition under some scenarios.

Strengths: Given the importance of goal/object recognition for decision-making in the presence of humans, it is surprising that enough work hasn’t been done to recognize the goals of bounded rational agents (given there is a lot of interest in modeling humans as bounded rational agents). I think the paper looks at an interesting problem and as far as I know, is the first paper to look at this specific setting at least for sequential decision-making problems and seems to be able to support settings with large state spaces.

Weaknesses: My main concerns for the work are about specific assumptions made regarding the agent's planning algorithm and how close the effectiveness of the goal recognition system is tied to having access to the specific planning algorithm and parameters used by the agent generating the observations. I would have liked to see experimental results that at least shows some level of robustness of the system towards mismatch between the planning algorithm used by the goal recognition system and the method used to generate the observations for the study. Below I have provided a more detailed discussion of my main concerns Specific Algorithm Used: The paper makes some specific assumptions on the kind of algorithm that could be used to simulate the bounded decision making. I see no reason to believe that this is general enough to capture behavior of any arbitrary resource bounded decision-maker (for example consider one that is quite similar to the one discussed, but is also memory bounded and can only hold limited possible nodes in its open list) or that this is in anyways similar to how a human would make such decisions (which is important if the primary goal is to be able to predict human goals). While the paper notes that people use heuristics as well, those may be quite different from the ones that are popular in planning literature. I can still imagine that even if the algorithms of the agent behaving and the one used by the recognizer doesn’t exactly match up, the goal recognition system may still be able to gather useful information. But this was never explicitly tested or established. I think a truly useful evaluation here would have been if there was some kind of user study to show that the system is able to correctly predict actual human goals. On a slightly smaller note, the original formulation of the problem seems to imply that the framework also supports problems with stochastic dynamics. But the actual planning system itself uses PDDL representations and uses A* search. Is the idea that the domain used by the planner is a determinization of the original stochastic domain or is the focus currently only on deterministic problems. I can see that the formulation can be extended to stochastic settings (determinization being one simple way), but I think it helps if all the assumptions used by the current system is explicitly noted upfront.

Correctness: I think all the theoretical claims are sound. The empirical study itself mostly focuses on the scalability of the system and soundness when the assumptions about planning matches up. I think a more interesting question could have been how robust this method would have been to variations in the actual agent’s decision-making model. At the very least the test could have used a different set of parameters for both observation generation and simulation during recognition and as far as I can see this isn't the case.

Clarity: I found the paper easy to read. As mentioned there seems to be some disconnect between the formulation and specific methods used in terms of the transition dynamics. Also I didn’t see details on whether the experiments were performed with noisy observation.

Relation to Prior Work: I think the work does a good job characterizing the works done in related space and contrasting with them.

Reproducibility: Yes

Additional Feedback: I would like the authors to respond to the following queries 1. Are there any reason to believe the system would be robust to scenarios where the agent may be using different decision making algorithms? 2. Is the current methods specifically designed for deterministic settings? 3. Did the evaluation use the exact same set of parameters for both generating the observation and for goal recognition? The paper currently only mention what was used to generate the observation. Post Rebuttal Comments: The additional results posted in the author response are definitely promising. I would urge the authors to run more experiments in this direction over more domains and more subjects. For the data collected from participants, it might be worth also noting whether you observed behavior that one might attribute to a bounded reasoner (like making mistakes, backtracking, etc..) or given the simple domains were they able to more or less come up with near-optimal plans.


Review 3

Summary and Contributions: -- Post rebuttal -- I have slightly raised my score to reflect the experiments regarding model robustness, but please make the discussion of model robustness an important point of the accepted paper. --- The paper considers goal inference (from other agent's trajectories) under an assumption that the trajectories were generated from an agent with bounded computational power. They approach the problem using probabilistic programming, inferring agents' goal by assuming to know their bounded rationality planning model.

Strengths: - The idea of looking at goal inference under a bounded rationality assumption is as far as I could tell novel and pretty interesting. - The overall approach is sensible and elegant if complex to implement. - The worked examples in figure 1 are very elucidating.

Weaknesses: - A major weakness of the paper is the circular way it is evaluated. Imperfect trajectories are generated using a particular model of goal-conditional planning bounded rationality. Then we perform inference on the goal of the agent assuming that the model of the agent is the exact same that generated the data. This is akin to generating random unsupervised data sets by a probabilistic program that generates random mixtures of gaussian, then perform model learning on these datasets, compare it to other baselines, and conclude that mixture of gaussians are exactly what you need for unsupervised learning. I don't object to the particular approach (we probably assume that others use a similar model of planning as we do), but to its evaluation. It seems clear that the generalisability / transfer of the inner model will be key to the usefulness of the overall model. If the mismatch is strong, the conclusions would likely be very different (say if data was generated from a depth-first search agent with bounded rationality, and the inference was done with a breadth-first agent). - For that matter, I am not terribly convinced by the model of bounded rationality adopted by the authors. It is strongly tied to a deterministic environment (a fact the authors acknowledge), and somewhat implies deciding one action at a time is suboptimal. We know from the theory of RL optimal actions can be computed without explicit planning (and even one-step planning can be optimal provided we do adequate bootstrapping). I understand the authors are going for a pure-planning, no heuristic/value functions approach, and so one-step planning is greedy, but this does not really reflect current state of the art planning method, nor does is it stated explicitly/discussed enough. - Finally, all planning is done under the assumption of a perfect model (goal aside), which should be more explicitly stated.

Correctness: Empirical methodology is flawed, see above.

Clarity: Yes, it is very clearly written.

Relation to Prior Work: Yes, it would valuable to cite [1] as well, which has a very similar approach, though more in multi-agent decision making than explicit goal inference. In [1], because the decision making process is a recursive probabilistic program, limited inference capacity (number of MCMC steps) and limited recursion depth naturally lead to another, perhaps more natural definition of bounded rationality. [1] Modeling Theory of Mind for Autonomous Agents with Probabilistic Programs, Seaman et. al

Reproducibility: Yes

Additional Feedback: In equation (8), the goal dependency needs to appear on the right hand side (otherwise posterior=prior). In section 5, it would make more sense to explain how data is generated before the baselines. --- post rebuttal: I have slightly risen my score given the experiments on human data addressing robustness to model mismatch.


Review 4

Summary and Contributions: The paper presents sequential inverse plan search (SIPS) as a novel method for online goal recognition. By assuming agents interleave planning and execution, SIPS avoids the cost of exhaustively computing all possible policies, while being robust to sub-optimal and failed plans. SIPS is also capable of handling partially observable states. SIPS behaves as a particle filter over the goal/plan space. Particles are updated as observations are received. SIPS is compared experimentally to Bayesian Inverse RL (BIRL), and outperforms BIRL in many cases.

Strengths: SIPS is a novel contribution providing an interesting extension on previous work. The presented algorithm is sound and based on well established particle filter methods. The work is grounded in terms of human decision making (ToM) and relevant to the NeurIPS community. The main text is supported by well designed and intuitive figures. The experimental results demonstrate the benefits of SIPS.

Weaknesses: The paper begins too technical, with unfamiliar concepts being discussed before being introduced (see remarks on clarity). The technical details of the rejuvenate function are unclear in the pseudocode and text. It is assumed that the agent’s planner is known, which is unrealistic in many settings (e.g. observing human behaviour). I think there is room for further experiments. For example, I would be interested to see the effects of an imperfect model of the agent’s planner.

Correctness: The extension of a particle filter for goal recognition is technically sound. The rejuvenation procedure is grounded in existing methods for improving particle diversity. The experimental methodology covers many appropriate domains, and provides hyperparameter values for reproducibility. I think the main text should further emphasise that SIPS can handle noisy/partial observations, as it demonstrates robustness.

Clarity: The paper is fairly well written. Pseudo code is well commented and mostly easy to follow. The main text is supported with excellent figures. However, the language is slightly too technical at the start. Certain concepts (e.g. boundedly-rational) are discussed prior to being introduced, and need a high-level description in the introduction. Further, the details of the rejuvenate function are unclear. The pseudo code overloads Q, adding confusion, and the alpha update (L31-32) for error-driven replanning is not explained.

Relation to Prior Work: SIPS expands on previous work by operating online, and proving to be robust to sub-optimal and failed plans. The related work covers all important areas, however in some cases the discussion is too brief. For example, there are many large blocks of citations, with no distinction made between the works cited.

Reproducibility: Yes

Additional Feedback: Minor Issues: 1. In Section 1, briefly describe plan recognition. 2. The caption of Fig. 1 should be cut from `In both cases…’. This is discussed in Section 5. 3. When introducing `boundedly rational` and `resource-limited plan search’ in Section 1, introduce that the bound is with respect to the number of planning steps. 4. Define the acronym SMC when Sequential Monte Carlo is first mentioned in Section 1. 5. In the related work, define a Boltzmann-rational MDP. 6. In 3.1, make it clear that the goal prior is from the observer’s perspective. 7. In 3.2 the negative in `disfavours small search budgets’ is difficult to parse. Use `favours longer search budgets’. 8. I think it may be easier to present the agent as using LRTDP for planning, which is similar to probabilistic A* but more well established in the literature. Similarly, discuss the effect of assuming an incorrect planner. 9. Typo in Eq. 8: P(p_t|s_t, p_t-1) -> P(p_t|s_t, p_t-1, g) 10. Relate Algorithm 1 to a particle filter. 11. Typo in Line 8, Algorithm 1: Resample([g -> Resample([g^i. 12. In Algorithm 1 line 24 it is unclear that BERNOULLI returns a sampled outcome. 13. Clarify how the new trajectory is sampled in Line 26 of Algorithm 1. Does it use the method in Lines 12-14? 14. Describe the effective sample size at a high level and with an equation. 15. Ignoring BIRL (oracle) when highlighting in bold in Table 1 is confusing. 16. The rejuvenation trade-offs discussed in the supplementary material should be in the main text. 17. The observation function definition should be included in the main text. 18. A source code link should be provided if possible. 19. When discussing interleaved planning and execution, refer to the phrase `receding horizon’. Questions: 1. Why does unbiased BIRL perform better in the Taxi environment? Why does BIRL (oracle) perform much better in some problems with respect to top-1? 2. How is the performance of SIPS affected by an imperfect model of the agent’s planner? For example, if r/q or T were incorrect, or we did not know the planner being used by the agent (e.g. if the agent is human)? 3. How well does SIPS perform in an offline setting when the agent does not replan at all?

[Author Response · NeurIPS 2020]

| Domain | Humans $n$=5 | RL Agents $\alpha$=50 | Persistence ($r$) | | | Persistence ($q$) | | | Search Temp. ($T$) | | | Heuristic ($h$) | | | |
| | | | 1 | 2* | 4 | 0.8 | 0.9 | 0.95* | 2 | 10* | 50 | Manhat.* | Maze.Dist. | Goal.Count. | $h_{\text{add}}$* |
| --- | --- | --- | --- | --- | --- | --- | --- | --- | --- | --- | --- | --- | --- | --- | --- |
| Doors, Keys, Gems | 0.79 | 0.58 | 0.60 | 0.73 | 0.73 | 0.53 | 0.60 | 0.73 | 0.67 | 0.73 | 0.77 | 0.73 | 0.90 | – | – |
| Block Words | 0.73 | 0.82 | 0.90 | 0.87 | 0.90 | 0.70 | 0.83 | 0.87 | 0.83 | 0.87 | 0.87 | – | – | 0.43 | 0.87 |

**Table 1: Robustness to model mismatch.** Top-1 accuracy of SIPS at the third time quartile (Q3), evaluated on data generated by humans, RL agents, and mismatched models. We ran SIPS assuming $r$=2, $q$=0.95, $T$=10, and a Manhattan ($h_{\text{add}}$) heuristic for Doors, Keys, Gems (Block Words). Matched parameters are starred (*).

We thank the reviewers for engaging carefully with our paper, and for providing helpful and constructive feedback. We
commit to addressing the minor issues raised and adding the suggested references.

**R1, R2, R3, R4 raised concerns about whether our model of boundedly-rational planning is robust to plans**
**generated by humans and other models**. We appreciate these concerns, and agree that it best to perform a user study
as R2 suggests, and to avoid the circular evaluation pointed out by R3. In response, we have performed a series of
robustness experiments (**Table 1**), showing that SIPS is robust to data from 5 pilot human subjects (30 trajectories per
subject), a Boltzmann RL agent, mismatched parameters $r$, $q$, $T$, and $h$. While performance can degrade with mismatch,
this is partly due to the difficulty of inference from highly random behavior (e.g. $q$=0.8, $h$=Goal Count). In fact, when
mismatched parameters are *more* optimal, performance can *improve* (e.g.$h$=Maze Dist.). Importantly, SIPS does well
on human data (Top-1=0.78 / 0.73), showing robustness even when the planner is unknown. **We have also conducted**
**preliminary experiments showing that human goal inferences correlate highly with SIPS** (Pearson's $r = 0.85$).
We will expand on these experiments in the final paper with more domains and cross-method comparisons.

**R1 asked about the motivation for using a negative binomial for search budgets** $\eta \sim \text{NB}(r, q)$. To explain, $r$ and
$q$ model the persistence of a planner who may give up after expanding each node. 1-$q$ is the probability that the planner
considers giving up, while $r$ is the number of times the planner has to consider giving up before actually giving up. This
captures the intuition that people are more likely to give up the longer we plan, while still exhibiting some persistence.
**R1 also expressed concerns about our relaxation of A\* search**, where a node $s$ is sampled from the open list for
expansion according to $P_{\text{expand}}(s) \propto \exp(-f(s, g)/T)$. To explain, this captures how agents may fail to choose the
best cognitive action (i.e. node expansion) during search, following a Boltzmann distribution. $T$ controls how informed
the search is, with $T$=0 (least informed) equivalent to breadth-first search, $T=\infty$ (most informed) equivalent to standard
A\*. Search is sound, because all nodes are eventually expanded, and path costs updated. Similar modifications are
made in both [**1**] and [**2**]. We will add these references in the final paper.

**R2 and R3 were also concerned whether our modeling assumptions are overly specific** *(R2. Q1, R3. Weaknesses)*.
We agree that there are many types of planning that go beyond the scope of this paper. As we note in Future Work,
this motivates extensions to domains like task and motion planning. To R3's point on optimal RL actions without
explicit planning, we agree this is a good model when humans are well-practiced. Still, when inferring plans for novel
tasks with sparse rewards, we believe explicit planning is a better model, and will gladly add discussion on this point.
**Planners can also exhibit many other bounds, including limited memory (R2) or inference capacity (R3).** We
see modeling such bounds as important future work. While our current model no doubt approximates human planning
by leaving out these bounds, humans are likely to use similarly approximate models when inferring the goals of others.
As **Table 1** shows, this approximation is enough to ensure reasonable robustness in the domains we consider.

**R1 asked about the importance of probabilistic programming for our research.** While R1 is right that the basic
particle filtering approach could be implemented manually with some additional work, our use of custom proposal
programs [**3**] and involutive MCMC [**4**] in the SIPS rejuvenation moves requires computing importance weight ratios for
all random choices in both the proposals and the model. Without Gen's automated weight computation, implementing
this would be highly tedious and error prone, akin to implementing back-propagation for a VAE without TensorFlow or
PyTorch. A manual implementation would also be much more difficult to extend to complex goal priors and planners.
We are happy to emphasize this in the final paper.

**R2 asked for clarification about whether our approach is tied to deterministic settings.** While all the evaluated
domains were deterministic, our framework also supports Probabilistic PDDL [**5**], and can be readily extended to
stochastic domains, e.g. by determinization as R2 suggests. Our use of a replanning model also means that it is not
as strongly tied to deterministic environments as R3 contends: if the agent encounters a state that it did not plan for
(e.g. randomly failing to pick up a block), it simply replans from that new state, which we believe to be a reasonable
approximation of human behavior in mostly deterministic domains. We will clarify these points in the final paper.

**R4 asked why unbiased BIRL performs better in the Taxi environment, and why oracle BIRL sometimes per-**
**forms better for top-1 accuracy.** This is because SIPS can suffer from particle collapse, whereas BIRL can perform
exact inference if it has good Q-value estimates (true for both oracle BIRL and the Taxi domain's small state space).
SIPS can be improved by increasing the particle count, using rejuvenation, or variance control techniques. We have
since implemented some of these techniques (e.g. residual resampling), and will update our results accordingly. **R4**
**also asked how well SIPS performs when the agent does not replan.** This is addressed in the supplement, where we
show that SIPS performs reasonably well on trajectories from an optimal full-horizon planner.

[Meta-Review · NeurIPS 2020]

This paper consider the task of inferring goals based on agent trajectories, under the assumption that trajectories are generated from an agent with bounded computational power. The authors model the agent using a probabilistic program that interleaves search and execution-by-replanning and perform inference using sequential inverse planning search (SIPS), an SMC method with rejuvenation. Reviewers overall leaned towards acceptance. The main concerns that reviewers raised centered on the fact that the proposed method assumes deterministic dynamics as well as well as knowledge of the planning strategy that is used by the agent. This leads to potential for model misspecification when, e.g., the true agent and the modeled agent employ a different planning algorithm. More broadly reviewers expressed some skepticism about the model of bounded rationality that is adopted in this approach. The authors provide some additional results in their author response that to an extent address concerns about model mismatch. The author response was positively received by reviewers, several of whom raised their scores, whilst providing a caveat that new results were somewhat difficult to evaluate given length constraints of the author response. The AC has carefully read the reviews, authors response, and discussion, and has also done a read-through of the submission. On balance the AC regards this paper as above the bar for acceptance, provided the authors take comments from reviewers regarding model misspecification and difficulty of model estimation to heart, and explicitly acknowledge these as limitations of the work in its current form.